# Primary Granulocyte Colony-Stimulating Factor Prophylaxis in Metastatic Pancreatic Cancer Patients Treated with FOLFIRINOX as the First-Line Treatment

**DOI:** 10.3390/cancers12113137

**Published:** 2020-10-27

**Authors:** Jae Hyup Jung, Dong Woo Shin, Jaihwan Kim, Jong-Chan Lee, Jin-Hyeok Hwang

**Affiliations:** 1Department of Internal Medicine, Seoul National University College of Medicine, Seoul National University Bundang Hospital, Seongnam 13620, Korea; 54279@snubh.org (J.H.J.); drjaihwan@snubh.org (J.K.); 2Division of Gastroenterology, Department of Internal Medicine, Keimyung University School of Medicine, Keimyung University Dongsan Medical Center, Daegu 42601, Korea; delight0618@dsmc.or.kr; 3Division of Gastroenterology, Department of Internal Medicine, Seoul National University Bundang Hospital 82, Gumi-ro 173 Beon-gil, Bundang-gu, Seongnam-si, Gyeonggi-do, Seongnam 13620, Korea

**Keywords:** pancreatic cancer, FOLFIRINOX, G-CSF prophylaxis

## Abstract

**Simple Summary:**

Because FOLFIRINOX shows high prevalence of hematologic toxicity, it is meaningful to investigate of usefulness of primary G-CSF prophylaxis in metastatic pancreatic cancer patients. In this retrospective study, a total of 165 patients were divided into G-CSF user group (*n* = 57) and non-user group (*n* = 105). Baseline characteristics were not significantly different between two groups, which included initial absolute neutrophil counts and metastatic burden. Primary G-CSF prophylaxis reduced the risk of neutropenia (55.6% to 31.6%, *p* = 0.003) and febrile neutropenia (18.5% to 1.8%, *p* = 0.002) and improved OS (8.8 to 14.7 months; hazard ratio (HR): 1.766, 95% CI: 1.257–2.481, *p* = 0.001). When administering FOLFIRINOX for MPC, primary G-CSF prophylaxis could be rationalized to reduced AEs and improve survival, and more prospective studies are needed.

**Abstract:**

Although FOLFIRINOX (5-fluorouracil, leucovorin, irinotecan, and oxaliplatin) has been proven efficacious in metastatic pancreatic cancer (MPC), physicians hesitate to administer it due to its hematologic toxicities. We investigated the usefulness of primary granulocyte colony-stimulating factor (G-CSF) prophylaxis. We reviewed electronic medical records of MPC patients with good performance status who were administered FOLFIRINOX as the first-line treatment from 2011 to 2017. The patients were divided into primary G-CSF prophylaxis users (group A) and non-users or therapeutic/secondary users (group B). Cumulative relative dose (cRDI), adverse effects (AEs), and overall survival (OS) were compared. A total of 165 patients (group A (57) vs. group B (108)) were investigated. Intergroup differences in baseline characteristics were not significant, although the cRDI and the number of treatment cycles were both higher in group A than in group B (cRDI: 80.6% vs. 73.9%, *p* = 0.007; 9 vs. 6 cycles, *p* = 0.004). Primary G-CSF prophylaxis reduced the risk of neutropenia (55.6% to 31.6%, *p* = 0.003) and febrile neutropenia (18.5% to 1.8%, *p* = 0.002) and improved OS (8.8 to 14.7 months; hazard ratio [HR]: 1.766, 95% CI: 1.257–2.481, *p* = 0.001). When administering FOLFIRINOX for MPC, primary G-CSF prophylaxis could be rationalized to reduced AEs and improve survival; more prospective studies are needed.

## 1. Introduction

Pancreatic cancer (PC) was the fourth leading cause of death from cancer in both men and women in the United States in 2019 [1]. Although combination chemotherapy regimens such as FOLFIRINOX (5-fluorouracil, leucovorin, irinotecan, and oxaliplatin) have considerably contributed to improvements in the overall survival (OS) of patients with metastatic PC (MPC) since 2011, the overall five-year survival rate is 9%, still indicating an unfavorable prognosis [2,3,4].

Since 2011, FOLFIRINOX is recommended as the first-line chemotherapy in MPC patients with good performance status [3,5]. Moreover, in 2019, modified FOLFIRINOX became a new a milestone in the adjuvant setting of resectable PC; consequently, the regimen is becoming more widely used [6]. However, FOLFIRINOX was demonstrated to show considerable hematologic and nonhematologic side effects, even with a dose-modified regimen. Therefore, modification of the FOLFIRINOX regimen while both maintaining efficacy and reducing toxicity remains an important issue [3,7,8,9,10,11,12,13].

In many retrospective studies, only 30–60% of PC patients who received FOLFIRINOX were administered primary granulocyte colony-stimulating factor (G-CSF) prophylaxis [14,15,16]. According to the National Comprehensive Cancer Network (NCCN) guidelines (version 2.2020) for the management of neutropenia, the risk group for febrile neutropenia among those treated with FOLFIRINOX for PC, changed from the intermediate-risk group to the high-risk group. However, routine primary G-CSF prophylaxis is still not recommended and is considered only in the high-risk population. Moreover, in some meta-analyses, primary G-CSF prophylaxis for solid tumors was associated with favorable OS [17,18,19,20,21]. 

Therefore, in the present study, we investigated the usefulness of primary G-CSF prophylaxis in reducing hematologic adverse effects (AEs) and improving survival in MPC patients with good performance status who were treated with FOLFIRINOX. 

## 2. Method 

### 2.1. Study Population 

We collected the data of all patients with histologically diagnosed pancreatic ductal adenocarcinoma from October 2011 to December 2017 at a single tertiary teaching hospital (Seoul National University Bundang Hospital, Seoungnam-si, Korea). Among the patients whose data were collected, MPC patients treated with FOLFIRINOX as the first-line treatment were retrospectively evaluated in the present study. Patients with a poor performance status (Eastern Cooperative Oncology Group (ECOG) performance status score of >2) were excluded. 

### 2.2. Study Design 

The patients were divided into two groups according to whether primary G-CSF prophylaxis was administered or not, namely, primary G-CSF prophylaxis users (group A) and non-users (group B). Patients for whom only secondary G-CSF prophylaxis and therapeutic G-CSF were used were included in the non-user group (group B). To investigate the safety and efficacy of FOLFIRINOX administered as the first-line therapy, the cumulative relative dose intensity (cRDI), AEs, and OS were compared by reviewing electronic medical records. The data cut-off date was 30 June 2020. cRDI was measured by considering dose reduction and chemotherapy interval modification between the starting day of chemotherapy and the day of the first radiological evaluation [22]. The occurrence of grade 3 or 4 AEs within three months after initiating chemotherapy was compared based on Common Terminology Criteria for Adverse Events v5.0. OS was defined as the time from the histological diagnosis of MPC to death or the last follow-up. Survival data were evaluated until 30 June 2020. 

### 2.3. G-CSF Agents

The G-CSF agent was used as filgrastim (subcutaneous, 300 mcg/day, three to five days) or pegfilgrasim (a PEGylated form of the filgrastim, subcutaneous, 6 mg per three weeks). The choice of two drugs was made according to the preference of the clinician, the requests of the patient, and the price.

### 2.4. Statistical Analysis

To compare the patients’ baseline characteristics, T-, chi-square, and Fisher’s exact tests were used. OS was evaluated using the Kaplan–Meier method and difference in survival was analyzed using the log-rank test. The Cox proportional hazard model was used to analyze survival and other factors. All tests were double-sided with a *p*-value of less than 0.05 for statistical significance. All analyses were performed using SPSS software version 22 (IBM corporation, New York, NY, USA).

### 2.5. Ethics Statement

The ethical approval for this study was obtained by the institutional review board of the Seoul National University Bundang Hospital, Seongnam, Korea, on 15 May 2020 (Approval Number B-2005/615-103). 

## 3. Results 

### 3.1. Characteristics of the Patients

A total of 165 patients (57 in group A vs. 108 in group B) met the eligibility criteria. Among them, 160 died during the follow-up period and three patients (one in group A vs. two in group B) remained alive until June 2020; the survival status of the remaining two patients (two in group A) was unknown. In group A, five patients were administered pegfilgrastim, while 52 were administered filgrastim.

The median age of the patients in both groups was 61.4 years; furthermore, 66.7% and 63.0% of the patients in groups A and B, respectively, were male. Intergroup differences (group A vs. group B) in pretreatment body mass index (BMI) (23.3 vs. 22.7 kg/m^2^, *p* = 0.147), serum albumin (4.1 vs. 3.9 g/dL, *p* = 0.544), and carbohydrate antigen (CA) 19-9 level (660.0 vs. 380.0 U/mL, *p* = 0.138) were not significant. The most common primary tumor site was the tail (52.6%) in group A and the head and neck (44.4%) in group B. Group A received a more intense FOLFIRINOX regimen (80.6% vs. 73.9% of cRDI, *p* = 0.007) and for a longer duration compared to those in group B (9.0 vs. 6.0 cycles, *p* = 0.004). Initial absolute neutrophil count (ANC) level was not different between the two groups (Table 1).

### 3.2. AEs 

Neutropenia and febrile neutropenia were more frequently observed in group B (31.6% vs. 55.6%, *p* = 0.003; 12.7% vs. 18.5%, *p* = 0.002, respectively), while the incidence of anemia and thrombocytopenia did not differ between the groups. In terms of nonhematologic toxicities, there were no significant intergroup differences in the incidences of fatigue, vomiting, diarrhea, and peripheral sensory neuropathy (Table 2). All patients who manifested neutropenic events (*n* = 60) received therapeutic G-CSF administration followed by secondary G-CSF prophylaxis.

### 3.3. Efficacy

Analyses using unadjusted variables showed that the administration of primary G-CSF prophylaxis (14.7 vs. 8.8 months; hazard ratio (HR) 1.766; 95% confidence interval (CI) 1.257–2.481) and normal albumin level (12.2 vs. 6.5 months; HR 1.864; 95% CI 1.295–2.683) exerted significant beneficial effects on OS. However, there was no significant OS benefit afforded by the use of higher dose intensity (≥76.3 of median cRDI) compared to that afforded by the lower dose (<76.3 of median cRDI) (11.6 vs. 9.5 months; HR 1.183, 95% CI 0.864–1.618). Old age, sex, and pretreatment CA 19-9 level were not identified as significant independent prognostic factors for OS. Analysis using adjusted variables showed that primary G-CSF prophylaxis (HR for death 1.799; 95% CI 1.249–2.591) and normal albumin level (HR for death 1.718; 95% CI 1.223–2.412) were independent favorable prognostic factors for OS (Table 3, Figure 1).

### 3.4. Primary G-CSF Prophylaxis and Clinical Outcomes by Age Group

Analysis of the treatment outcomes according to age groups (non-older patients [age < 65 years] vs. older patients (age ≥ 65 years)) showed that primary G-CSF prophylaxis exerted a significant beneficial effect on OS (14.5 vs. 8.7 months; HR 1.957, 95% CI 1.258–3.042), while reducing the risk of neutropenia (38.9% vs. 60.6%, *p* = 0.036) and febrile neutropenia (0.0% vs. 16.7%, *p* = 0.010) and increasing cRDI (85.2% vs. 76.4%, *p* = 0.007) in non-older patients (Appendix A, Figure 2A). However, primary G-CSF prophylaxis did not have a significant effect on OS (14.7 vs. 8.8 months; HR 1.519; 95% CI 0.878–2.629), while reducing the risk of neutropenia (19.0% vs. 47.6%, *p* = 0.028) and not significantly reducing the risk of febrile neutropenia (4.8% vs. 21.4%, *p* = 0.088) with similar cRDI (72.2% vs. 70.1%, *p* = 0.334) in the older patient group. (Appendix A, Figure 2B).

## 4. Discussion

In the present study, we attempted to investigate the routine use of primary G-CSF prophylaxis as a strategy for reduction of hematologic AEs and thus potentiation of the efficacy of FOLFIRINOX. We evaluated 165 patients with MPC, which was a relatively large number, and found that only 35.0% of the patients were administered primary G-CSF prophylaxis. Our results demonstrated that primary G-CSF prophylaxis reduced hematologic AEs (e.g., neutropenia and febrile neutropenia), and that an increase in dose intensity and treatment duration of FOLFIRINOX conferred a survival benefit in MPC patients. Similar to previous studies [17,18,19,20,21], our study demonstrated that routine primary G-CSF prophylaxis use is justified given its beneficial effects (considering the reduction in AEs in conjunction with the survival benefit).

Since it was recommended in 2011, FOLFIRINOX has considerably contributed to improving the survival of MPC patients [2,3,4,5,23]. Several retrospective and prospective studies showed that AEs due to FOLFIRINOX were more frequently observed compared to in the study by Conroy et al. [5,7,8,24,25]. Version 2.2020 of the NCCN guidelines emphasizes the risk of neutropenic fever associated with FOLFIRINOX administration in PC patients. Therefore, dose-modified regimens were studied in vulnerable patients; however, the efficacy of these regimens remains controversial [9,10,11,12,13].

According to the NCCN guidelines version 2.2019, FOLFIRINOX use is recommended for intermediate-risk patients for febrile neutropenia, and the administration of primary G-CSF prophylaxis may be recommended in patients aged 65 years or higher when FOLFIRINOX is prescribed at full-dose intensity. This recommendation was based on a randomized controlled trial conducted by Conroy et al., which showed that the rate of febrile neutropenia was only 5.4% in the FOLFIRINOX group (G-CSF was administered to 42.5% of the patients) [5]. However, several retrospective studies reported that febrile neutropenia was more frequently observed (14% in KOREAN study [22] and 22.2% in a Japanese trial [24]) than reported in the randomized trial by Conroy et al. (5.4%) in MPC patients who received FOLFIRINOX [5]. Although sufficient evidence was lacking, FOLFIRINOX was recommended for high-risk individuals for febrile neutropenia (febrile neutropenia risk of >20%) in the NCCN guidelines version 2.2020. Despite this revised risk stratification, routine primary G-CSF prophylaxis is not recommended for all populations and is only currently considered in the high-risk population. Our findings indicate that routine prophylactic G-CSF use might be more beneficial than expected. 

In our study, we performed a subgroup analysis by age (<65 years vs. ≥65 years). Primary G-CSF prophylaxis reduced neutropenia in both subgroups (60.6% to 38.9% vs. 47.6% to 19.0%, respectively). Contrary to expectations, primary G-CSF prophylaxis reduced febrile neutropenia in only the non-older group (16.7% to 0.0%), although the older group showed a tendency toward febrile neutropenia reduction associated with primary G-CSF prophylaxis (21.4% to 4.8%). Furthermore, primary G-CSF prophylaxis conferred a significant survival benefit in only the non-older group, although the trend was similar in both groups, possibly because the number of older patients was too small to draw a conclusion. Another explanation could be that primary G-CSF prophylaxis had a favorable effect on survival as dose intensity was increased in the non-older group. In the older group, however, regardless of the administration of prophylactic G-CSF, the FOLFIRINOX dose intensity was similar. This perhaps reflects physicians’ reluctance to initiate FOLFIRINOX at full-dose intensity because of concerns about AEs, despite the administration of primary G-CSF prophylaxis. Further studies are warranted to confirm these issues. 

In the previous study, we found that greater dose reduction is needed in elderly patients with advanced pancreatic cancer when using FOLFIRINBOX [22], as reflected in Appendix A. Since it was mentioned in previous studies that excessive dose reduction affects tumor response [22], it can be hypothesized that cRDI may also affect OS, which could be the subject of subsequent studies.

This study has several limitations. There is a possibility of selection bias because the patients were from a single tertiary teaching hospital. Furthermore, because there was no blinding, there is a possibility of performance bias by the physicians. Furthermore, this study did not consider the patients’ socioeconomic status, which determines whether a patient receives primary G-CSF prophylaxis or not as the intervention is not reimbursed by national insurance in South Korea. Nevertheless, because the median patient age at diagnosis of PC has increased over the past 10 years (65 to 70 years), we believe that our results may be valuable when making a decision (primary G-CSF prophylaxis vs. a wait-and-see strategy) in older MPC patients treated with FOLFIRINOX.

## 5. Conclusions

In conclusion, primary G-CSF prophylaxis could improve survival when FOLFIRINOX is administered in MPC patients while reducing hematologic AEs, allowing for an increase in dose intensity and treatment duration of FOLFIRINOX. Thus, routine primary G-CSF prophylaxis could be justified in this patient population. More well-designed, prospective studies are necessary for large populations.

## Figures and Tables

**Figure 1 cancers-12-03137-f001:**
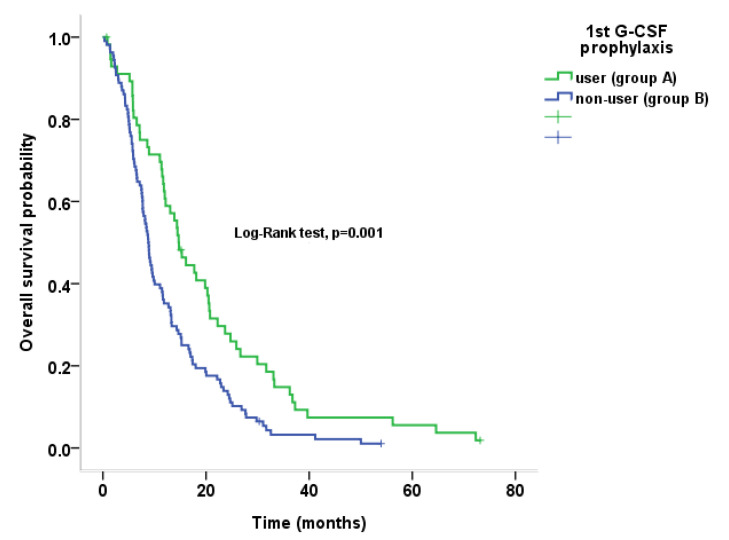
Overall survival in primary G-CSF users and non-users.

**Figure 2 cancers-12-03137-f002:**
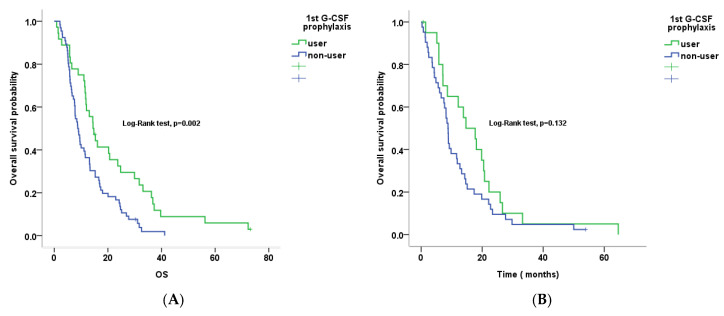
Overall survival in primary G-CSF users and non-users ((**A**) non-older patient subgroup; (**B**) older patient subgroup).

**Table 1 cancers-12-03137-t001:** Baseline characteristics.

Variables	1st G-CSF User (*n* = 57)	1st G-CSF Non-User (*n* = 108)	All Patients (*n* = 165)	*p* Value
Age, median (range, yr)	61.1 (41–80)	61.5 (29–79)	61.4 (29–80)	0.856
Sex, no. (%)				0.637
Female	19 (33.3)	40 (37.0)	59 (35.8)	
Male	38 (66.7)	68 (63.0)	106 (64.2)	
BMI, median (range, kg/m^2^)	23.3 (15.4–29.1)	22.7 (15.0–29.3)	23.0 (15.0–29.3)	0.147
ECOG performance status				0.095
0–1	51 (89.5)	104 (96.3)	155 (93.9)	
2	6 (10.5)	4 (3.7)	10 (6.1)	
Primary site, no. (%)				0.098
Head and neck	20 (35.1)	48 (44.4)	68 (41.2)	
Body	5 (8.8)	19 (17.6)	24 (14.5)	
Tail	30 (52.6)	39 (36.1)	69 (41.8)	
Multicentric	1 (1.8)	2 (1.9)	3 (1.8)	
Site of metastasis, no. (%) *				0.874
Liver	38 (66.7)	67 (62.0)	105 (63.6)	
Peritoneal seeding	13 (22.8)	26 (24.1)	39 (23.6)	
Distant lymph node	17 (29.8)	21 (19.4)	38 (23.0)	
Lung	12 (21.1)	20 (18.5)	32 (19.4)	
Others	13 (22.8)	21 (19.4)	34 (20.6)	
Metastatic organ, no. (%)				0.391
1	32 (56.1)	72 (66.7)	107 (64.8)	
2	15 (26.3)	23 (21.3)	37 (22.4)	
≥3	10 (17.5)	13 (12.0)	21 (13.3)	
Baseline laboratory values				
ANC, median (range, g/dL)	4624 (840–11,297)	4723 (1,331–12,178)	4712(840–12,178)	0.642
Albumin, median (range, g/dL)	4.1 (2.5–4.7)	3.9 (2.4–5.0)	3.9 (2.4–5.0)	0.544
CA19-9, median (range, U/mL)	660.0 (5.0–20,000)	380.0 (5–20,000)	464.6 (5–20,000)	0.138
cRDI, median (range)	80.6 (43.3–102.2)	73.9 (28.3–109.0)	76.3 (28.3–109.0)	0.007
Cycles, median (range)	9.0 (1–75)	6.0 (1–31)	7.0 (1–75)	0.004

Data are presented as median (range) or number of patients/total number (*n*%), unless otherwise stated. * Site of metastasis includes multiorgan involvement. G-CSF, granulocyte colony-stimulating factor; BMI, body mass index; ECOG, Eastern Cooperative Oncology Group; ANC, absolute neutrophil count; CA 19-9, carbohydrate antigen; cRDI, cumulative relative dose intensity; cRDI = (actual cumulative dose/standard cumulative dose) × 100.

**Table 2 cancers-12-03137-t002:** Serious adverse effect within three months.

Grade 3–4 Toxicity	1st G-CSF User (*n* = 57)	1st G-CSF Non-User (*n* = 108)	All Patients (*n* = 165)	*p* Value
Hematologic				
Neutropenia	18/57 (31.6)	60/108 (55.6)	78/165 (47.3)	0.003
Febrile neutropenia	1/57 (1.8)	20/108 (18.5)	21/165 (12.7)	0.002
Anemia	3/57 (5.3)	17/108 (15.7)	20/165 (12.1)	0.050
Thrombocytopenia	2/57 (3.5)	9/108 (8.3)	11/165 (6.7)	0.237
Nonhematologic				
Fatigue	1/56 (1.8)	5/105 (4.8)	6/161 (3.7)	0.342
Vomiting	3/56 (5.4)	13/105 (12.4)	16/161 (9.9)	0.156
Diarrhea	5/56 (8.9)	7/105 (6.7)	12/161 (7.5)	0.603
Sensory neuropathy	0/56 (0.0)	2/105 (1.9)	2/161 (1.2)	0.299

Data are presented as number of patients/total number (%), unless otherwise stated. Grade 3–4 according to the Common Terminology Criteria for Adverse Events (CTCAE), version 4.03.

**Table 3 cancers-12-03137-t003:** Unadjusted and adjusted relationships between administration of G-CSF with other variables and overall survival.

Variables	Number of Patients (%)	OS (Median, Months)	95%CI (Months)	Unadjusted			Adjusted		
				HR	95%CI	*p* Value	HR	95% CI	*p* Value
Overall patients	165	11.0	9.0–13.1						
Age									
<65 yr	102 (62)	11.4	9.3–13.5						
≥65 yr	63 (38)	9.2	6.1–12.3	1.154	0.837–1.591	0.381			
Sex									
Female	59 (36)	11.6	7.8–15.3						
Male	106 (64)	9.5	6.6–12.3	1.297	0.936–1.797	0.118			
Albumin									
≥3.5 g/dL	124 (75)	12.2	10.3–14.0						
<3.5 g/dL	41 (25)	6.5	4.3–8.7	1.864	1.295–2.683	0.001	1.718	1.223–2.412	0.002
CA 19-9									
<464.6 U/mL	78 (50)	13.0	10.9–15.2						
≥464.6 U/mL	78 (50)	8.9	8.2–9.6	1.166	0.845–1.607	0.350			
Primary G-CSF prophylaxis									
Yes	57 (35)	14.7	12.0–17.4						
No	108 (65)	8.8	7.9–9.7	1.766	1.257–2.481	0.001	1.799	1.249–2.591	0.002
Dose intensity (cRDI)									
≥76.3	83 (50)	11.6	8.1–15.2						
<76.3	82 (50)	9.5	7.1–11.8	1.183	0.864–1.618	0.295			

Data are presented as median (range) or number of patients (%), unless otherwise stated. OS denotes overall survival; CI, confidence interval; ECOG PS, Eastern Cooperative Oncology Group Performance Status; cRDI, cumulative relative dose intensity, cRDI = (actual cumulative dose/standard cumulative dose) × 100.

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
