# Peer review of "Primary Granulocyte Colony-Stimulating Factor Prophylaxis in Metastatic Pancreatic Cancer Patients Treated with FOLFIRINOX as the First-Line Treatment"

_cancers, 2020, doi:10.3390/cancers12113137_

Round 1

Reviewer 1 Report

This retrospective cohort study examined the effects of G-CSF prophylactic treatment on adverse events and overall survival among pancreatic cancer patients treated at a tertiary medical center.

The main critique involves the extent to which the patients in group A (G-CSF users) differ from patients in group B (non-users), beyond the few variables reported in Table 1. Baseline characteristics at the time of diagnosis may inform a physicians’ decision to prescribe G-CSF. Are there additional clinicopathologic characteristics (e.g., number/sites of metastases) or relevant lab test results (e.g., baseline neutrophil count) available in the EHR data that the authors can include in Table 1 to evaluate differences between patients who receive G-CSF and those who didn’t?

The authors should consider performing propensity score matching (on propensity of receiving G-CSF) as a sensitivity analysis to the findings reported, which will help guard against any biases due to factors at baseline influencing the likelihood of receiving G-CSF that are also associated with the outcome.

Why did the authors decide to include patients who received secondary G-CSF prophylaxis and therapeutic G-CSF in the non-user group? How many patients fell into this sub-category? Wouldn’t this potentially have an impact on the occurrences of adverse events or OS?

Lin 94 – Define pegfilgrastim and filgrastim in the introduction or methods, as the first time they are used is in the results without any background information on what they are.

Line 117 and Table 3 – Univariate analysis means the description of one variable, not the relationship between two variables. The authors should consider rephrasing this section/table as reporting unadjusted and adjusted relationships between administration of G-CSF (and other study variables) and survival.

Author Response

We attach a word file. Thank you.

Reviewer 2 Report

This is a well written study that covers a relevant topic. Previous studies have come to similar conclusions, but many studies on the inclusion of G-CSF with FOLFIRINOX are extremely underpowered. The current study covers 165 patients.  

Minor Concerns:

  1. The authors could include a statement about a prospective study being necessary in order to truly understand the clinical benefit
  2. Some of the abbreviations need to be spelled out prior to usage
  3. The authors also need to address the fact that the cRDI was significantly increased in the <65 age group in the G-CSF arm and the interpretation of increased OS in this group

Author Response

We attach a word file. Thank you.

Reviewer 3 Report

The Authors retrospectively analyze the use of G-CSF prophylaxis to prevent adverse effects linked to FOLFIRINOX treatment in metastatic pancreatic cancer patients. While data support reduction of hematological side effects upon G-CSF primary administration, authors also report improvement in overall survival.

Of note, I am not an epidemiologist, not l have particular expertise in evaluating large cohort studies. To me, the study is neat and presents no major issues.

While the authors touch a little bit on that, I think there is a possibility that a selection bias explains the better OS in the A group (G-CSF+). Can the authors comment a little bit more on why some patients (a smaller group, retrospectively) received G-CSF and others (the majority) did not? Authors cannot rule out the possibility that patients in group A had better prognostic indicators to begin with, had better access to care or even had a less severe disease. To that point, group A has an abnormal percentage of tumors located in the tail of the pancreas. I really don't see reasons to ascribe the improved survival to G-CSF administration. The authors should simply report the (very interesting) association observed.

Maybe the authors can add more infos on the disease staging at the beginning of therapy?

The one suggestion I would make is to tune down the conclusion of the abstract. Administration of G-CSF seems an interesting option, but additional studies, including prospective studies, are necessary before advocating for its regular use, in my view.

Author Response

We attach a word file. Thank you.

Round 2

Reviewer 1 Report

All comments have been addressed. No additional critiques.